# STATEFUL MULTI-AGENT EVOLUTIONARY SEARCH FOR UNIT TEST GENERATION

## ABSTRACT

Recent work explores agentic inference-time techniques to perform structured, multi-step reasoning. However, stateless inference often struggles on multi-step tasks due to the absence of persistent state. Moreover, task-specific fine-tuning or instruction-tuning often achieve surface-level code generation but remain brittle on tasks requiring deeper reasoning and long-horizon dependencies. To address these limitations, we propose **stateful multi-agent evolutionary search**, a training-free framework that departs from prior stateless approaches by combining (i) persistent inference-time state, (ii) adversarial mutation, and (iii) evolutionary preservation. We demonstrate its effectiveness in automated unit test generation through the generation of edge cases. We generate robust edge cases using an evolutionary search process, where specialized agents sequentially propose, mutate, and score candidates. A controller maintains persistent state across generations, while evolutionary preservation ensures diversity and exploration across all possible cases. This yields a generalist agent capable of discovering robust, high-coverage edge cases across unseen codebases. Experiments show our stateful multi-agent inference framework achieves substantial gains in coverage over stateless single-step baselines, evaluated on prevalent unit-testing benchmarks such as HumanEval and TestGenEvalMini and using three diverse LLM families—Llama, Gemma, and GPT. These results indicate that combining persistent inference-time state with evolutionary search materially improves unit-test generation.

## 1 INTRODUCTION

Despite their success on single-step tasks, most inference-time computation in large language models (LLMs) remains stateless, with each inference call discarding prior intermediate reasoning unless explicitly re-injected into the prompt. This design choice optimizes deployment throughput but cripples performance in domains that require deep, multi-stage reasoning—such as program synthesis, theorem proving, multi-hop reasoning, deductive reasoning, and mathematical problem-solving—where intermediate states must be persistently updated and revisited. The fixed computational depth per transformer forward pass (Vaswani et al., 2017) and the well-documented decline in reasoning fidelity over long logical chains (Wei et al., 2022; Anil et al., 2022) make these limitations structural rather than incidental. The autoregressive decoding process further constrains exploration by forcing reasoning branches to unfold serially, often necessitating brittle orchestration through multiple model calls (Yao et al., 2023a; Long et al., 2024). Overcoming these constraints demands stateful inference-time architectures—including scratchpad prompting (Nye et al., 2021; Wei et al., 2022), tree-structured reasoning (Yao et al., 2023b), and retrieval-augmented agents (Lewis et al., 2020)—that can maintain and manipulate intermediate reasoning artifacts directly. However, current techniques still operate by eliciting reasoning from fixed, opaque model parameters, limiting both steerability (Zhou et al., 2023) and interpretability (Olah et al., 2020; Nanda et al., 2023) of the reasoning process.

In this context, we investigate the suitability of a multi-stage evolutionary algorithm (Bäck et al., 1997; Hansen, 2016) in which each stage executes an adversarially guided actor–critic-style (AGAC) search (Ding et al., 2023). Unlike conventional evolutionary pipelines where each generation is evaluated in isolation, our design shares state information—captured as the critic's value estimates—across successive evolutionary stages. This shared evaluation signal serves as a persistent knowledge base, allowing later stages to inherit and refine the judgment of earlier stages rather

than re-learning from scratch (Jaderberg et al., 2017; Such et al., 2017). Such cross-stage information flow improves sample efficiency, reduces evaluation variance, and encourages coherent policy evolution over long optimization horizons (Mouret & Clune, 2015; Salimans et al., 2017). By integrating AGAC within this multi-stage framework, we can combine the exploration benefits of evolutionary search with the fine-grained feedback of actor–critic learning (Konda & Tsitsiklis, 2000). In this work, the actor–critic terminology strictly refers to inference-time reward shaping only: the critic scores candidate tests to guide the actor, but no model parameters are updated as in traditional reinforcement learning.

This distinction is important because our aim is not to train new policies, but to adapt inference-time behavior for practical tasks. Unit test generation provides an ideal setting to study this: it requires structured reasoning beyond syntax, benefits directly from persistent state, and offers measurable signals such as coverage and mutation scores to guide search. Stateless test generation often covers only a narrow slice of behavior, whereas maintaining state enables gradually expanding coverage and surfacing deeper failure modes. In summary, this work introduces a training-free framework for unit test generation that (i) maintains persistent inference-time state across search iterations, (ii) integrates coverage, exceptions, and mutation robustness into a unified reward design, and (iii) demonstrates consistent coverage improvements over stateless baselines on HumanEval and Test-GenEvalMini.

## 2 RELATED WORK

The rapid advancement of large language models (LLMs) has enabled significant progress in AI-assisted reasoning, code generation, and test automation. Prior research spans several domains including code generation, test synthesis, algorithmic discovery, and scientific reasoning, yet many approaches face limitations in adaptability, coverage, and generalization.

Multi-agent frameworks such as AI Co-scientist (Gottweis et al., 2025), AlphaEvolve (Novikov et al., 2025b) and GEPA (Agrawal et al., 2025) demonstrate that collaborative reasoning and reflective prompt evolution can enhance exploration. However, these systems typically lack persistent state and rely on ad-hoc orchestration rather than structured reward signals.

Evolutionary and search-based methods preserve high-fitness candidates and explore combinatorial program behaviors Mühlenbein et al., 1988; Burnim & Sen, 2008. In particular, evolutionary search explicitly manages the exploration-exploitation tradeoff, allowing the system to explore novel program behaviors while retaining high-performing edge cases and avoiding local minima that static or greedy methods often encounter. Other works Karten et al.; Leng et al., 2024; Wen et al., 2024 provide insights into scaling, planning, and iterative hypothesis generation, emphasizing structured search and evaluation.

Despite these advances, prior approaches often struggle with adaptability, robust coverage, and systematic exploration of edge cases. Many LLM-based test generators operate in a feed-forward manner or require fine-tuning, limiting their ability to dynamically adjust to new or evolving codebases. Multi-agent and evolutionary approaches in prior work may fail to explore the full combinatorial space or integrate adversarial evaluation effectively.

We address these gaps with a training-free framework that unifies (i) multi-agent reasoning, (ii) adversarial mutation and reward shaping, and (iii) evolutionary preservation with persistent state. An actor proposes candidate edge cases by reasoning over the program, an adversary perturbs the code to expose hidden failure modes, and a critic integrates coverage, exceptions, and mutation feedback to prioritize high-value test cases. A non-Markovian controller maintains memory of prior edge cases and preserves high-fitness candidates across iterations, enabling inference-time policy adaptation and robust exploration. This combination allows our system to dynamically adapt to unseen codebases, produce robust edge cases, and achieve higher coverage than existing methods, without relying on gradient-based training or domain-specific fine-tuning.

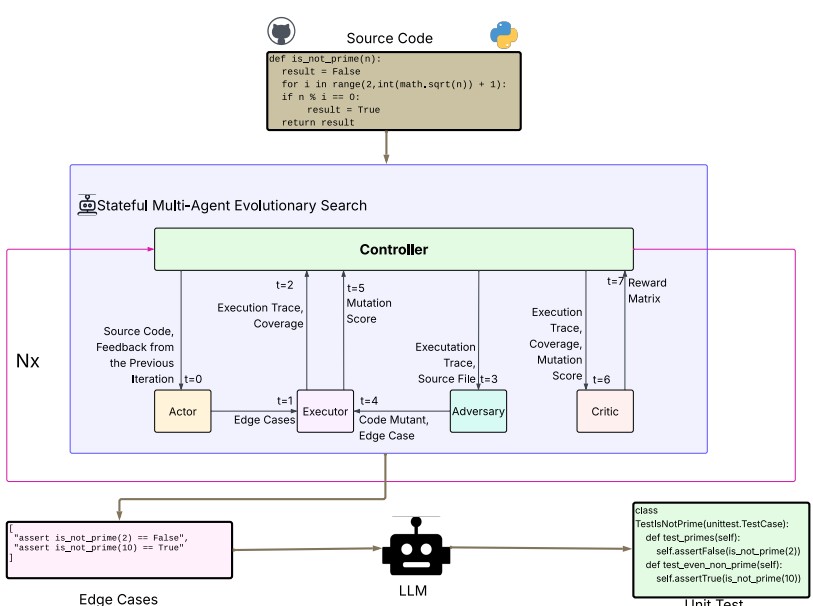

Figure 1: Our architecture for unit test generation decomposes the task into two phases: (i) edge case generation from source code and (ii) unit test construction from those cases. The first phase demands deeper reasoning and is addressed through an evolutionary search (as highlighted in the blue box) executed in a stateful manner over multiple stages ($N\times$) by four agents—Actor, Executor, Adversary, and Critic—coordinated by a Controller that propagates persistent state across $N$ evolutionary stages (as highlighted by the magenta line). Once the edge cases converge to sufficient coverage and robustness, they are translated into a complete unit test file via a single-step inference call.

## 3 METHODOLOGY

Our central premise is that generating syntactically correct unit tests is trivial once a set of robust edge cases with sufficient coverage are identified, but reasoning about such edge cases requires structured exploration, memory, and adversarial grounding.

Figure 1 shows the architecture for the unit test generation engine with the proposed *stateful multi-agent evolutionary search* for the edge case generator. Given source code $f$, the system first runs the stateful multi-agent evolutionary search to extract edge cases and then converts those cases into unit tests.

Our stateful multi-agent evolutionary search is an adversarially guided actor-critic (AGAC) system that operates entirely at inference time and does not require gradient-based learning. The **Actor** issues multiple LLM inference calls to propose candidate edge cases, the **Adversary** perturbs the environment to reveal robustness gaps, and the **Critic** assigns scalar rewards used for evolutionary search. The **Executor** is an auxiliary agent that provides an execution environment to execute edge cases, unit tests, and return coverage and robustness feedback. These four agents are orchestrated through the **Controller** which maintains persistent state across stages and orchestrates the search until convergence.

**Definition 1** (State). A State is represented in Equation 1,

$$S_{n-1} = \Big( \zeta_{1:n-1}, \mu_{1:n-1}, \kappa_{1:n-1}, c_{1:n-1}, R_{1:n-1} \Big) \tag{1}$$

where $\zeta_{1:n-1}$ denotes the sequence of prior edge cases, $\mu_{1:n-1}$ is the sequence of mutation scores, $\kappa_{1:n-1}$ is the sequence of coverage scores, $c_{1:n-1}$ is the sequence of exception signals, and $R_{1:n-1}$ is the reward history from previous stages.

**Definition 2** (Actor). The Actor ($A_n$) proposes candidate edge cases at each stage. At initialization ($n = 1$), there is no prior feedback or state information to guide generation, so the actor is seeded deterministically (cold-start) using rule-based heuristics such as boundary partition analysis, equivalence classes, and stress conditions. For $n > 1$, the actor generates candidates through large language model in-context learning, conditioned on the persistent state $S_{n-1}$ and the source code $f$:

$$\zeta_n = \mathcal{A}(f, S_{n-1}) \tag{2}$$

More details about the cold-start can be found in Appendix D

**Definition 3** (Adversary). For each stage, Adversary ($D_n$) generates a set of mutants $\{f'_{n,j}\}_{j=1}^M$ of the source file, and evaluates whether the edge cases $\zeta_n$ can kill these mutants (i.e produce a different output on $f'_{n,j}$ than they did on $f$). The resulting mutation score is defined in Equation 3 and provides a robustness signal for evaluating the edge case candidates. Mutation testing promotes robustness by checking whether tests can distinguish the true program from systematically perturbed variants, preventing the search from optimizing toward shallow coverage gains.

$$\mu_n = \frac{\text{Number of mutants killed by } \zeta_n}{\text{Total number of generated mutants}} \tag{3}$$

**Definition 4** (Critic). For each stage, Critic ($\mathcal{C}_n$) computes the scalar reward for the edge cases by integrating coverage ($\kappa$), mutation robustness ($\mu$), and exception discovery ($c$), given by Equation 4.

$$R_n^{\text{unnormalized}}(\kappa_n, \mu_n, c_n) = [\alpha \cdot c_n + \beta(\kappa_n + \max(0, (\kappa_n - \theta) \cdot 0.5))] \times \gamma \cdot \mu_n \tag{4}$$

where $\alpha, \beta, \theta, \gamma \in \mathbb{R}_+$ are tunable hyperparameters. All rewards are normalized to $[0, 1]$ using min-max normalization for evolutionary comparison.

The reward combines exception discovery ($c_n$), structural coverage ($\kappa_n$), and mutation robustness ($\mu_n$). The exception term encourages exploration of inputs that expose faults. The coverage term accounts for the proportion of program elements exercised, with an additional bonus once a minimum threshold $\theta$ is passed, so that progress beyond trivial coverage is reflected more strongly. Multiplication by the mutation score ensures that high reward is assigned only when the generated tests are also robust to program perturbations. By shaping the critic's reward surface using adversarial perturbations we ground the actor's responses and thus prevent the actor from optimizing toward trivial coverage gains instead of exploring robust, high-value edge cases.

**Definition 5** (Executor). All evaluations for coverage and mutation scoring are executed in a sandboxed Docker environment with a Model-Context Protocol (MCP) server. This provides: (i) **Isolation:** Mutants and edge cases cannot harm the host system; (ii) **Determinism:** Results are reproducible across runs; and (iii) **Bounded resources:** Memory and timeouts prevent unbounded execution. A detailed description of the Executor architecture can be found in Appendix A.3.

**Definition 6** (Controller). The controller orchestrates the interplay of Actor, Adversary, and Critic by updating the non-Markovian state information (Equation 1) and checking for the termination criteria as defined in Equation 5.

$$\sum_i R_i \geq \tau \quad \text{or} \quad \max_{i \in [n-p+1,\, n]} R_i \; - \; \min_{i \in [n-p+1,\, n]} R_i \leq \delta \tag{5}$$

The controller applies two complementary stopping conditions. The first checks whether the reward has crossed a predefined threshold, indicating that the search has reached a sufficient overall quality level. The second detects a plateau in rewards over the most recent $p$ iterations, suggesting that further search is unlikely to yield substantial improvements. The plateau condition is evaluated only when $n \geq p$. The thresholds ($\tau, \delta$) and window size $p$ can be tuned according to task complexity as well as computational budget, allowing the framework to balance thoroughness and efficiency.

A key methodological contribution is that our framework does not require training, fine-tuning, or task-specific adaptation of large language models. Instead, it builds a training-free test-generation agent whose intelligence emerges from:

1. **Inference-time state management:** The controller maintains a structured non-Markovian state, feeding the actor with explicit histories of edge cases, coverage scores, mutation feedback, and exceptions. Unlike conventional RL where state updates drive gradient descent, here state updates directly shape the actor's inference context. This functions as a lightweight form of policy shaping at inference time, guided by rewards but without parameter updates.

2. **Multi-agent grounding:** The actor's outputs are consistently grounded by adversarial mutations and fitness evaluation from the critic, allowing even base LLMs without domain adaptation to be repurposed into reasoning agents.

3. **Evolutionary selection:** The framework preserves a population of diverse elites, avoiding reliance on a single trajectory and improving robustness without requiring specialized training.

This positions our framework alongside recent agentic paradigms such as AI Co-Scientist (Gottweis et al., 2025) and AlphaEvolve (Novikov et al., 2025a), while differing in its explicit use of evolutionary preservation and adversarial reward shaping to structure inference-time coordination. Algorithm 1 shows the overall computational framework that we use for multi-stage evolutionary search.

## 4 EXPERIMENTS

We evaluated the proposed evolutionary search algorithm on two benchmark datasets, HumanEval and TestGenEvalMini, using three large language models (LLMs): `Llama-70B`, `GPT-o4-mini`, and `Gemma-2-27B`. To assess its effectiveness, we compared our method against six inference-time baselines—zero-shot, one-shot, and three-shot in-context learning, each with and without chain-of-thought (CoT) prompting—under three standard test coverage metrics: line coverage, branch coverage, and function coverage. We use coverage.py for line/branch/function metrics and Cosmic-Ray for mutation analysis. Each run is sandboxed in a Docker/MCP environment.

**HumanEval:** HumanEval Chen et al., 2021 is a benchmark of 164 Python programming problems with reference implementations. HumanEval is designed to test reasoning and correctness in code generation. For evaluation, all examples were typeset for consistency and compatibility with automated execution frameworks, allowing precise assessment of model outputs, including edge-case handling and exception detection.

**TestGenEvalMini:** Derived from the original TestGenEval dataset Zhang et al., 2024 (which is built from SWEBench Jimenez et al., 2024), TestGenEvalLite contains real-world code and test file pairs from 11 well-maintained Python repositories. TestGenEvalLite preserves the complexity of real-world software engineering, including multi-parameter interactions, boundary conditions, and exception handling. The dataset was reformatted and type-annotated for structured evaluation and automated execution. TestGenEvalLite is a benchmark released for unit test generation tasks on repositories which preserve the complexity of real-world software engineering. TestGenEvalMini is a curated subset of TestGenEvalLite containing 48 representative examples across 6 repositories, intended for rapid experimentation in constrained execution environments. Modules that trigger multiple MCP requests in rapid succession (e.g., Django autoreload) or require complex cross-functional dependencies were excluded to ensure stability. This mini benchmark allows researchers to rapidly test the effectiveness of edge-case reasoning and test generation techniques in a controlled environment before scaling to larger datasets. Importantly, while TestGenEvalMini reduces setup overhead, our static analysis (Table 1) shows that its structural complexity remains comparable to TestGenEvalLite. Code length, number of functions, and branching constructs span a similar range, ensuring that TestGenEvalMini provides a representative challenge for model evaluation while being optimized for fast execution.

**Dataset Contribution:** We release curated versions of both HumanEval and TestGenEvalMini, augmented with detailed edge-case traces containing coverage, mutation, and exception metadata. These traces enable the fine-tuning or training of reasoning models without requiring full-scale program execution. The resulting datasets span use cases from rapid prototyping to large-scale eval-

---

**Algorithm 1** Adversarially Guided Actor–Critic with Evolutionary Search for Unit Test Generation

---

**Require:** Source file $f$
**Ensure:** Final Unit Test File UT

1: Initialize $n \leftarrow 1$, $S_0 \leftarrow \emptyset$, $R_0 \leftarrow 0$
2: **while** not ShouldStop($\{R_1, \ldots, R_{n-1}\}, n - 1$) **do**
3:   **Actor:**

$$\zeta_n = \begin{cases} A(f) & n = 1 \quad \text{(cold start: rule-based heuristics)} \\ A(f, S_{n-1}) & n > 1 \end{cases}$$

4:   **Executor:** Run $\zeta_n$ on $f$ to obtain execution results $\rho_n$ and coverage $\kappa_n$
5:   **Adversary:** Mutate $f$ into $\{f'_{n,1}, \ldots, f'_{n,M}\}$, execute $\zeta_n$, and compute

$$\mu_n = \frac{K_n}{K_n + S_n}$$

  where $K_n$ and $S_n$ are killed and survived mutants.
6:   **Executor:** Compute exception signals $c_n = \text{ExceptionSignal}(\rho_n)$
7:   **Critic:** Compute reward

$$R_n^{\text{unnorm}}(\kappa_n, \mu_n, c_n) = [\alpha \cdot c_n + \beta(\kappa_n + \max(0, (\kappa_n - \theta) \cdot 0.5))] \times \gamma \cdot \mu_n$$

$$R_n = \frac{R_n^{\text{unnorm}} - R_{\min}}{R_{\max} - R_{\min}}$$

8:   Update archive: retain top-$K$ edge cases from $\zeta_{1..n}$ by reward $R_n$

$$\zeta_{1:n} \leftarrow \text{top-}K\big(\zeta_{1:n}, \text{sorted by } R_{1:n}\big)$$

9:   Set $n \leftarrow n + 1$
10:   Update state:

$$S_n = (\zeta_{1..n}, \mu_{1..n}, \kappa_{1..n}, c_{1..n}, R_{1..n})$$

11: **end while**
12: **Synthesis:** UT $\leftarrow$ LLM($f, S_n$)
13: **return** UT
14:
15: **Function ShouldStop**($\{R_1, \ldots, R_n\}, m$):

$$\textbf{if } m < p \textbf{ then return } \Big( \sum_{i=1}^{m} R_i \geq \tau \Big)$$

$$\textbf{else return } \Big( \sum_{i=1}^{m} R_i \geq \tau \Big) \vee \Big( \max_{i \in [m-p+1,\, m]} R_i - \min_{i \in [m-p+1,\, m]} R_i \leq \delta \Big)$$

---

| Metric | Lite (160 tasks, 11 repositories) | Mini (48 tasks, 6 repositories) |
|---|---|---|
| Code LOC | $906.57 \pm 821.67$, median = 584 | $575.79 \pm 600.78$, median = 425 |
| Functions | $46.27 \pm 53.80$, median = 31 | $33.81 \pm 37.38$, median = 28 |
| Branches | $79.87 \pm 84.46$, median = 52 | $60.06 \pm 70.57$, median = 40 |

Table 1: Comparison of structural complexity metrics between TestGenEvalLite and TestGenEvalMini.

uation, thereby supporting reproducible research in inference-time agentic reasoning for software testing.

Table 2: Final Edge Case Quality for HumanEval for Llama 70B

| HumanEval | Line Coverage | Branch Coverage | Function Coverage |
|---|---|---|---|
| SUT | 90.01% | 89.76% | 91.51% |
| Zero Shot LLM | 82.77% | 81.92% | 85.36% |
| Zero Shot LLM with CoT | 86.90% | 86.73% | 87.5% |
| One Shot LLM | **90.85%** | **90.70%** | **92.07%** |
| One Shot LLM with CoT | 87.21% | 87.04% | 88.41% |
| Three Shot LLM | 89.94% | 89.87% | 90.09% |
| Three Shot LLM with CoT | 88.18% | 88.13% | 89.33% |

Table 3: Final Unit Test File Quality for TestGenEvalMini

| TestGenEvalMini | Line Coverage | Branch Coverage | Function Coverage |
|---|---|---|---|
| SUT Llama 70B | **29.80%** | 16.55% | **29.24%** |
| SUT o4-mini | 28.22% | 15.28% | 27.78% |
| SUT Gemma-2-27B | 26.95% | 14.88% | 28.05% |
| Zero Shot LLM | 22.59% | 15.45% | 24.62% |
| Zero Shot LLM with CoT | 22.31% | 16.02% | 22.83% |
| One Shot LLM | 25.22% | 14.95% | 26.58% |
| One Shot LLM with CoT | 25.24% | 15.22% | 27.28% |
| Three Shot LLM | 25.35% | **17.40%** | 26.83% |
| Three Shot LLM with CoT | 24.66% | 16.21% | 25.80% |

## 5 RESULTS

### 5.1 HUMANEVAL

HumanEval consists of standalone, file-level implementations, where the relative advantage of advanced inference-time strategies is inherently limited. As shown in Table 2, the system-under test (SUT) and all six inference-time baselines perform comparably, serving as a sanity check for our proposed evolutionary search method. Our evolutionary search method achieves comparable final edge case quality while requiring zero additional LLM calls in approximately 62% of cases. This highlights that the *cold-start* stage of our system is powerful: seeded by deterministic heuristics such as boundary partitioning and equivalence classes, it often produces high-quality edge cases without requiring iterative refinement. Thus, HumanEval problems collapse almost entirely at initialization, demonstrating both the efficiency of our framework and the need for stronger benchmarks such as TestGenEvalMini to highlight the benefits of multi-agent evolutionary reasoning.

### 5.2 TESTGENEVALMINI

Figure 2 reports the final edge case quality achieved by our evolutionary search method compared to six inference-time baselines. With `Llama-70B`, our approach consistently outperforms all baselines by substantial margins across line, branch, and function coverage. In contrast, this trend weakens for `GPT-o4-mini` and `Gemma-2-27B`: the system under test (SUT) continues to achieve the highest line and function coverage, but is surpassed by these models in branch coverage. This discrepancy may stem from a tendency of our search to emphasize exception-heavy or assert-focused tests, which can thoroughly exercise one control-flow path without necessarily exploring its complements. While this bias lowers measured branch coverage, it often surfaces deeper failure modes that line and function metrics capture. We view this as a promising avenue for future refinement, where incorporating branch-aware objectives could balance thorough path exploration with the strong exception discovery our method already provides. Remarkably, there is little difference between the few-shot settings with and without chain-of-thought (CoT) prompting, both in terms of coverage metrics and the number of inference calls required, highlighting the need for stateful mechanisms to achieve reasoning without post-training.

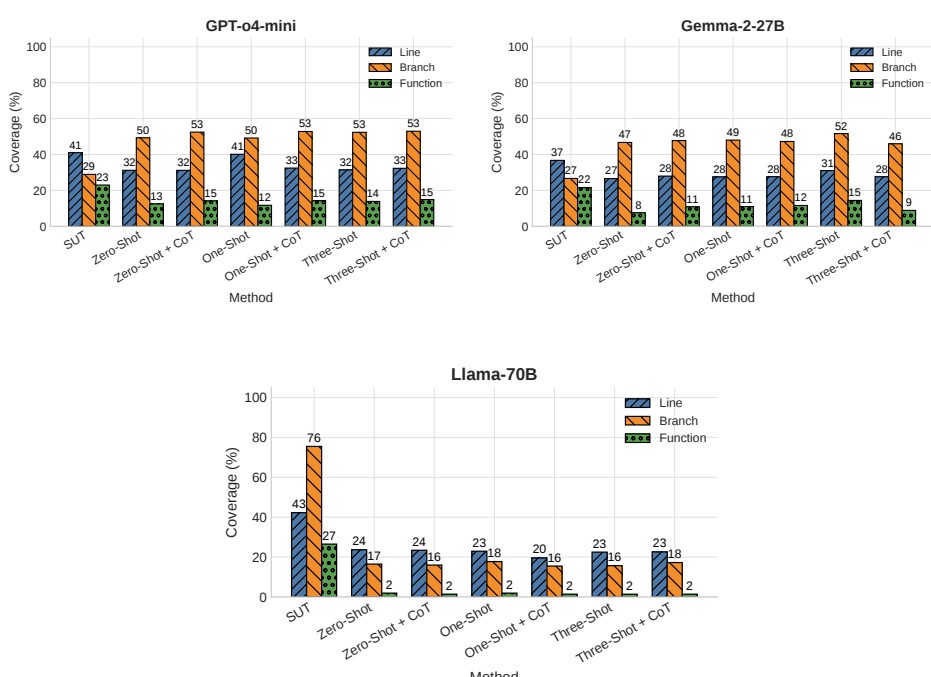

Figure 2: Final edge case quality on TESTGENEVALMINI measured in terms of line, branch, and function coverages across three model families: GEMMA-2-27B (top-left), GPT-O4-MINI (top-right), and LLAMA-70B (bottom). The proposed inference-time evolutionary search (SUT) consistently achieves strong coverage, outperforming few-shot and chain-of-thought baselines in most settings.

Figure 3 presents the resolution rate (blue, left axis) and average runtime (red, right axis) for two benchmarks. The **resolution rate** is defined as the fraction of generated unit tests that successfully reach convergence. In the left subplot, HUMANEVAL shows that nearly $62\%$ of problems are resolved in a single iteration, with only modest runtime overhead, indicating that the majority of tasks are relatively straightforward. In contrast, the right subplot for TESTGENEVALMINI exhibits a markedly different profile: while the majority of problems require three or more iterations, resolution rates plateau only after extended search, with runtimes rising steeply at higher iteration counts. Together, these results highlight the efficiency of our inference-time evolutionary search on simpler benchmarks, while also demonstrating its ability to scale to more complex tasks at the cost of additional compute. The prompts can be found in Appendix B and an example unit test file generation can be found in Appendix C.

Overall, our evolutionary search achieves higher coverage than inference-time baselines across HumanEval and our TestGenEvalMini. While branch coverage lags slightly for GPT-o4-mini and Gemma-2-27B, this likely reflects differences in how these models explore control-flow paths; refining branch-focused operators is an avenue for future work. For efficiency, we report representative runs, and the patterns we observe are stable across models and subsets.

## 6  CONCLUSION

We introduced a stateful multi-agent evolutionary framework for unit test generation, which departs from stateless inference by maintaining persistent reasoning state across multiple stages of search. By combining an actor for edge-case proposal, an adversary for robustness evaluation, a critic for reward integration, and an executor for sandboxed verification, our system achieves substantial gains in coverage compared to few-shot and chain-of-thought baselines. Experiments on HumanEval and TestGenEvalMini demonstrate that stateful evolutionary search enables higher coverage edge-case discovery, scaling beyond the capabilities of conventional stateless prompting. These results high-

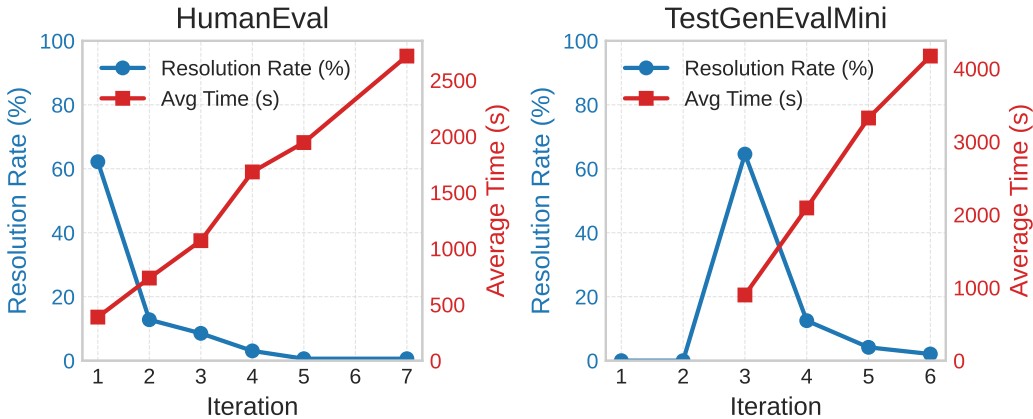

Figure 3: Evolution of line coverage over iterations for the three model families. LLAMA-70B improves over about four stages before stabilizing, while GPT-O4-MINI and GEMMA-2-27B plateau earlier.

light the promise of inference-time multi-agent coordination as a training-free strategy for improving the reasoning depth and reliability of large language models.

Nonetheless, several limitations remain. The proposed stateful multi-agent evolutionary framework incurs higher inference-time compute costs and longer runtimes on complex tasks, potentially limiting deployment in latency-sensitive settings. Future work will focus on extending the executor to handle richer dependency contexts, developing more efficient search termination criteria, and incorporating learned reward models to stabilize scoring. Broader evaluation across multilingual benchmarks and industrial-scale repositories will also be critical to assess generalization. Addressing these challenges will enable more practical, scalable, and adaptive inference-time agents for automated software testing.

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

# A APPENDIX

## A.1 COMPUTATION COST (FLOPs)

We report floating-point operation counts (FLOPs) for a single evaluation **iteration** of our adaptive pipeline. FLOPs provide a hardware-agnostic measure of computational demand and allow princi-pled comparison across model sizes and ablations.

We decompose the iteration into language-model (LLM) calls and non-LLM procedures (code ex-ecution, mutation, bookkeeping). For autoregressive transformer inference, we adopt the standard accounting

$$\text{FLOPS}_{\text{LLM}} \approx 2\mathcal{N}_{\text{params}} \cdot \mathcal{T}$$

where $\mathcal{N}_{\text{params}}$ is the number of model parameters and $\mathcal{T}$ is the total number of tokens processed (prompt + generated). The factor 2 reflects the dominant matrix multiplications in the forward pass. (If back-propagation were involved, a factor $\approx 3x$ the forward cost would be appropriate; our pipeline uses inference only.)

Non-LLM components are counted analytically from primitive operations in the relevant procedures (e.g., parsing, AST transforms, interpreter startup), yielding FLOPs that are negligible relative to LLM usage but included for completeness.

A.2  FLOPs FORMULATION

We derive the total floating point operations (FLOPs) required per iteration of our Actor–Adversary–Critic loop. Let:

- $N_{\text{actor}}$: number of parameters in the Actor LLM
- $N_{\text{ut}}$: number of parameters in the UnitTest LLM
- $L_{\text{src}}$: source code length (tokens)
- $R$: number of rule variations (edge cases) generated per iteration
- $R_{\text{ut}}$: maximum number of edge cases to keep for unittest generation
- $M$: max number of mutants (code mutations) executed per iteration
- $T_{\text{others}}$: average tokens of system prompt, task description etc
- $T_{\text{ec}}$: average tokens per edge case description
- $T_{\text{ut\_out}}$: output length of the generated unit test suite (tokens)
- $F_{\text{exec}}$: FLOPs per code execution
- $F_{\text{mut}}$: FLOPs per mutation generation
- $F_{\text{critic}}$: FLOPs per critic evaluation
- $F_{\text{other}}$: FLOPs for JSON parsing, string processing, and logging

**1. Actor FLOPs.**  The Actor LLM processes both source code and accumulated context to generate new edge cases.

$$T_{\text{actor\_in}} = L_{\text{src}} + (R \cdot T_{\text{ec}}) + T_{\text{others}}$$

$$T_{\text{actor\_out}} = R \cdot T_{\text{ec}}$$

$$T_{\text{actor}} = T_{\text{actor\_in}} + T_{\text{actor\_out}}$$

$$F_{\text{actor}} = 2 \cdot N_{\text{actor}} \cdot T_{\text{actor}}$$

**2. Unittest FLOPs.**  If the system makes use of an LLM to generate the final unittest file as opposed to a Human in the Loop, then these computations also need to be taken into account.
The UnitTest LLM consumes the source and filtered edge cases to produce complete test suites.

$$T_{\text{ut\_in}} = L_{\text{src}} + (R_{\text{ut}} \cdot T_{\text{ec}}) + T_{\text{others}}$$

$$T_{\text{ut}} = T_{\text{ut\_in}} + T_{\text{ut\_out}}$$

$$F_{\text{ut}} = 2 \cdot N_{\text{ut}} \cdot T_{\text{ut}}$$

**3. Code Execution FLOPs.**  Each generated mutant and the original source are executed against all edge cases.

$$E = (M + 1) \cdot R$$

$$F_{\text{exec\_total}} = E \cdot F_{\text{exec}}$$

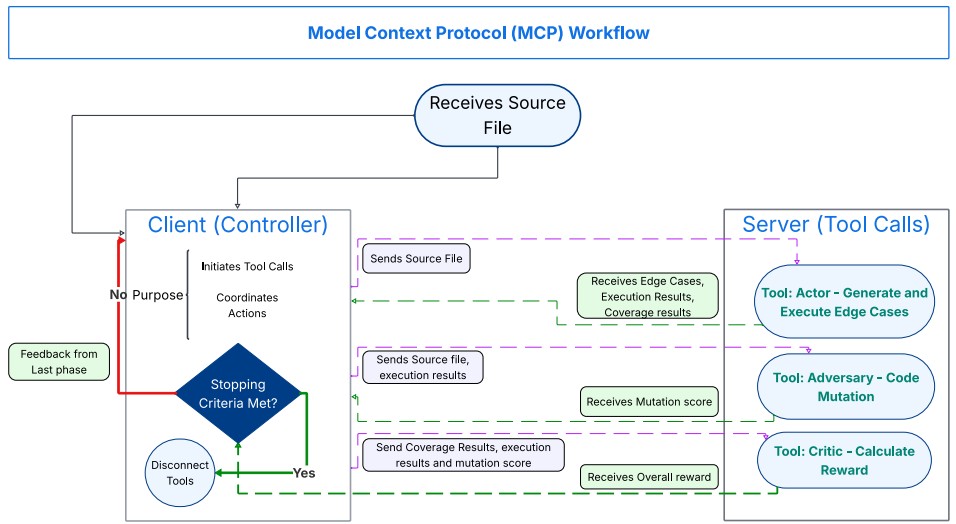

Figure 4: MCP Architecture overview

**4. Mutation FLOPs.**  Considering an average of 30 mutants are generated in every iteration (after which $M$ are randomly sampled for execution).

$$F_{\text{mut\_total}} = 30 \cdot F_{\text{mut}}$$

**5. Critic and Other FLOPs.**

$$F_{\text{critic\_total}} = R \cdot F_{\text{critic}}$$

$$F_{\text{other\_total}} = F_{\text{other}}$$

**6. Total System FLOPs.**

$$F_{\text{system}} = F_{\text{actor}} + F_{\text{ut}} + F_{\text{exec\_total}} + F_{\text{mut\_total}} + F_{\text{critic\_total}} + F_{\text{other\_total}}$$

This formulation allows us to compute FLOPs analytically for different evaluation settings, such as TESTGENEVALMINI and HUMANEVAL, by substituting the corresponding parameter values.

Running the system on TestGenEvalMini requires **3584.0 TFLOPs** per LLM Iteration, and an additional **819.2 TFLOPs** for the final unit test file generation. The TFLOPs for all other computation are negligible, including rule-based generation (which only requires an average of **36000 FLOPs**. Running the system on HumanEval requires **812.0 TFLOPs** per LLM Iteration, and an additional **128.0 TFLOPs** for the final unit test file generation. The TFLOPs for all other computation are negligible, including rule-based generation (which only requires an average of **13500 FLOPs**).

| Category | TestGenEvalMini (TFLOPs) | HumanEval (TFLOPs) |
|---|---|---|
| LLM Iteration | 3584.0 | 812.0 |
| Final Unit Test Generation | 819.2 | 128.0 |
| Rule-based / Other Computation | 0.036 | 0.0135 |

## A.3  EXECUTOR

The ***Executor*** is an integral auxiliary component within our system architecture that facilitates the *Controller* in managing the orchestrated flow of information. It employs a Model Context Protocol (MCP) Client-Server framework to ensure secure and isolated execution of all generated edge cases and mutated code variants. To maintain strict isolation, all executions on the server side are containerized using Docker, thereby sandboxing them from the host environment.

### A.3.1 MCP WORKFLOW

The operational workflow of the Executor is depicted in Fig. 4 and proceeds as follows:

1. The Executor receives the input source file for testing.

2. The Client Controller coordinates the process and initiates invocations of the various MCP tools.

3. The source file is transmitted from the client to the MCP Server through a tool call directed to the *Actor*.

4. The Actor module generates pertinent edge cases and executes them on the source file within the sandboxed environment.

5. The MCP Server returns the generated edge cases, execution outcomes, and coverage metrics to the client.

6. The client forwards both the source file and execution results to the MCP Server through a tool call to the *Adversary*.

7. The Adversary produces mutations of the source code and runs the previously generated edge cases on these mutants, again within the sandboxed environment, ultimately computing a mutation score.

8. This mutation score is returned from the MCP Server to the client.

9. Subsequently, the client transmits the execution results, coverage data, and mutation score to the MCP Server via a tool call to the *Critic*.

10. The Critic aggregates this information to compute a comprehensive reward, which it then returns to the client.

11. Finally, the Client Controller evaluates predefined stopping criteria:
    - If the criteria are satisfied, the tools are cleanly disconnected.
    - Otherwise, all feedback generated during the current rollout is assimilated and forwarded, along with the source file, back to the Actor to initiate the subsequent rollout.

### A.3.2 LIMITATIONS OF THE EXECUTOR

Despite its current capabilities, the Executor exhibits several limitations:

1. The system presently supports only single source files and lacks comprehensive repository indexing, thereby limiting its ability to handle dependencies spanning multiple files or relative package imports.

2. Certain file types, particularly those that return complex serialized objects (e.g., pickled files), are not currently supported.

3. Modules that initiate multiple MCP requests in quick succession, such as Django's autoreload module, may cause server instability and disconnections.

4. Dependency extraction is automated using `pipreqs`; however, unresolved version mismatches and dependency conflicts occasionally arise, which `pipreqs` cannot resolve.

These limitations necessitate the exclusion of such cases in the present implementation. Nonetheless, we anticipate that with a more sophisticated Executor design, our adversarially guided Actor-Critic framework can be extended to generate tests for these more complex scenarios using the established MCP workflow. Enhancing the Executor environment will thus substantially increase the robustness and applicability of the overall architecture.

## B PROMPTS

### B.1 EDGE CASE REASONING PROMPT

#### B.1.1 LLM EDGE CASES SYSTEM PROMPT

```
def llm_edge_cases_system_prompt():
```

```
      return """
      ### ROLE ###
      You are the **ACTOR** in an Actor{Adversary{Critic (AAC) loop
      for automated code testing.

      - **Actor (you):** Generate diverse, high-value test cases to maximize code
      coverage and detect edge failures.
      - **Adversary:** Mutates inputs to find weaknesses.
      - **Critic:** Scores inputs based on coverage, exceptions,
      and semantic boundaries.

      ### MISSION ###
      Generate **new**, **distinct**, and **high-impact** edge cases for
      *all* given functions.

      ### METHODS ###
      Use techniques including Boundary Value Analysis, Equivalence Partitioning,
      Scenario Testing, Random Testing, Stress Testing, Exception Triggering,
      and Complex Multi-parameter Interactions.

      ### OUTPUT FORMAT ###
      - Output **valid JSON only** in this exact format:
        `{ "function_name": [ { "param1": value, ... }, ... ] }`
      - Keys must be function names; values are arrays of parameter dictionaries.
      - Values must be valid JSON literals only (number, string, boolean, null,
          array, object).
      - **Do NOT include any explanatory text or formatting outside of JSON.**
      - **Do NOT include JavaScript expressions or comments.**

      ### FEEDBACK INTEGRATION ###
      - Incorporate the provided feedback to improve and diversify edge cases.
      - Avoid repeating previously generated edge cases.
      - Ensure new cases target untested or under-tested scenarios.
      """

def llm_edge_cases_system_prompt_with_cot():
    base_prompt = llm_edge_cases_system_prompt()
    cot_addition = """
        ### REASONING INSTRUCTIONS ###
        Before generating edge cases, carefully analyze the feedback,
        especially focusing on:

        - **Maximizing line coverage:** Identify uncovered or poorly covered lines
        in the source code.
        - Uncovered branches and exceptions not yet triggered.
        - Parameters or code paths with low test coverage.

        Think step-by-step about how to design new test cases that specifically
        target these uncovered lines to increase overall coverage.

        **Important:** Do NOT include your reasoning in the final output.
        Output **only valid JSON** edge cases that reflect this reasoning.

        """
    return base_prompt + cot_addition
```

### B.1.2 LLM EDGE CASES USER PROMPT

```
def llm_rule_expander_prompt(
```

```
810     function_signatures: Dict[str, List[str]],
811     source_code: str,
812     feedback_summary: str,
813     edge_cases_generated: str,
814     target_count: int
815 ) -> str:
816     """
817     Generate prompt for LLM to create edge cases for ALL functions at once
818
819     Args:
820         function_signatures: Dict mapping function names to their parameter lists
821         source_code: The complete source code
822         feedback_summary: Feedback from adversary/critic
823         edge_cases_generated: Previously generated edge cases
824         target_count: Total number of edge cases to generate across all functions
825     """
826
827     # Format function signatures for the prompt
828     ... code not included for brevity...
829
830     functions_list = "\n".join(functions_info)
831
832     prompt = f"""
833         SRC CODE:
834         ```
835         {source_code}
836         ```
837         FUNCTIONS:
838         {functions_list}
839
840         FEEDBACK FROM LAST RUN:
841         {feedback_summary}
842
843         TASK:
844         Generate {target_count} NEW and DISTINCT edge cases distributed
845         **evenly across all functions** above.
846
847         GUIDANCE:
848         - Incorporate all feedback to improve coverage and trigger new exceptions.
849         - Do NOT repeat previous edge cases.
850         - Generate valid JSON ONLY | strictly adhere to the output format.
851         - Focus on edge, boundary, and rare case inputs.
852         - Distribute edge cases fairly across functions.
853         - Provide no text outside the JSON.
854
855         OUTPUT EXAMPLE:
856         {{
857         "function1": [
858             {{"param1": "value1", "param2": 0}},
859             {{"param1": "value2", "param2": -1}}
860         ],
861         "function2": [
862             {{"x": 999999, "y": -999999}},
863             {{"x": 0, "y": 0}}
864         ]
865         }}
866
867         GENERATE JSON ONLY.
868     """
```

## B.2 FINAL EDGE CASES TO UNIT TEST FILE GENERATION PROMPT

### B.2.1 LLM UNIT TEST GENERATION SYSTEM PROMPT

```python
def edge_cases_to_unittest_system_prompt():
    return """
    You are an expert Python test generator.
    Your task is to convert the given edge cases **and doctest/typical examples**
    into pytest unit tests.

    RULES:
    1. Output ONLY valid Python 3.11 code
    | no markdown, no explanations, no extra text.
    2. Use EXACTLY 4 spaces per indentation level (no tabs).
    3. All parentheses, brackets, and braces must be balanced.
    4. Import only pytest and built-ins if needed | no extra imports.
    5. Each edge case must become one complete pytest test function.
    6. Test names must follow: test_<function>_<short_scenario>.
    7. Use literals exactly as shown (Ellipsis → ..., Infinity → float("inf"), etc.).
    8. Function parameters and variables MUST be valid Python identifiers:
        – Must start with a letter or underscore
        – May contain letters, numbers, or underscores
        – Must NOT start with a digit (incorrect: "3_14" → correct: "val_3_14")
    8a. If the edge case uses unclear or undefined variables
    (e.g., threshold_3_14, Array_1000_0):
        – Replace them with safe, concrete Python literals:
            – Numbers: 0, 1, 3.14
            – Lists: [], [0], [None] as appropriate
            – Strings: '', 'example'
            – Objects: None
    9. Edge case handling:
        – {"input": {...}, "expected": X} → assert function output == X
        – {"input": {...}, "raises": "ExceptionType"}
        → wrap call in pytest.raises(ExceptionType)
        – {"input": {...}} only → just call the function
    9b. For **normal/typical inputs** (including doctests),
    generate pytest functions with **assert statements** for expected results.
    10. Avoid duplicates: if multiple edge cases are semantically identical,
    merge them into one test function.
    11. Every generated test file must pass a syntax check:
        `python -m py_compile generated_tests.py`
    12. Mentally simulate importing and running the file to confirm:
        – All tests execute without NameError, TypeError, SyntaxError,
        or undefined variables.
    13. Always include at least one test for **valid input with assert**,
    even if edge cases exist.
    14. Convert all doctest-style examples (>>> lines)
    into pytest assert statements.
    15. Do NOT invent new literals or variable names; always use safe defaults
    if input is unclear.

    DO NOT OUTPUT ANYTHING OTHER THAN THE TEST CODE.
```

### B.2.2 LLM UNIT TEST GENERATION USER PROMPT

```python
def edge_cases_to_unittest_prompt(
```

```
        source_code: str,
        edge_cases,  # Can be list of dicts or JSON string
) -> str:
    # Handle both list of edge cases and JSON string
    import json
    if isinstance(edge_cases, str):
        edge_cases_repr = edge_cases
    else:
        # Convert list of edge cases to formatted JSON
        edge_cases_repr = repr(edge_cases)
    prompt = f"""
Convert the following edge cases into a complete pytest test file.

    SOURCE CODE:
    {source_code}

    EDGE CASES (JSON):
    {edge_cases_repr}

    REQUIREMENTS:
    - One pytest test function per edge case.
    - Use the schema rules from system prompt
    (expected → assert, raises → pytest.raises).
    - Ensure all test functions are syntactically correct and executable.
    - Absolutely no invalid parameter names (e.g., those starting with digits).
    - Convert all doctest-style examples (>>> lines)
    into pytest assert statements.
    - For error cases, use pytest.raises to assert the correct exception is raised.
    - Ensure a good mix of assert and pytest.raises statements.

    Now generate the pytest test file:
    """
    return prompt
```

## B.3    BASELINES EDGE CASE REASONING PROMPT

### B.3.1    BASELINES LLM EDGE CASES SYSTEM PROMPT

```
    def edge_case_generation_system_prompt():
    return """
    You are a Python expert. Your job is to generate **diverse,
    high-value edge cases** for given functions.

    CRITICAL RULES:
    1. Output ONLY valid JSON | no explanations, markdown, or extra text.
    2. Format must be strictly:
       { "function_name": [ { "param1": value, ... }, ... ] }
    3. Keys = function names, Values = arrays of input dictionaries.
    4. JSON literals only: number, string, boolean, null, array, object.
    5. Use Boundary Value Analysis, Equivalence Partitioning, Exception Triggering,
    Stress Testing, and Unusual Combinations.

    FORMATTING REQUIREMENTS:
    - Start your response with { and end with }
    - Use double quotes for all strings and keys
    - Do NOT include any text before or after the JSON
    - Do NOT wrap the JSON in markdown code blocks
    - Ensure all brackets and braces are properly balanced
    - Each function must have at least one edge case
```

```
972          - Parameter values must be valid JSON types
973          (no Python-specific values like None, True, False
974          - use null, true, false instead)
975          """
976
977
```

### B.3.2   BASELINES LLM EDGE CASES USER PROMPTS

```
979   def edge_case_generation_user_prompt(
980       source_code: str,
981       function_signatures: Dict[str, List[str]],
982       extra_text: str = "",
983       cot_flag: bool = False
984   ) -> str:
985       # Format function signatures for clarity
986       functions_info = []
987       for func_name, params in function_signatures.items():
988           if params:
989               functions_info.append(f"  - {func_name}({', '.join(params)})")
990           else:
991               functions_info.append(f"  - {func_name}()")
992       functions_list = "\n".join(functions_info)
993       #function_signatures_json = json.dumps(function_signatures, indent=2)
994
995       prompt = f"""
996   {extra_text}
997
998   SOURCE CODE:
999   {source_code}
1000
1001  FUNCTIONS TO TARGET:
1002  {functions_list}
1003
1004  TASK:
1005  Generate new, distinct, and high-impact edge cases for all listed functions.
1006
1007  OUTPUT FORMAT:
1008  {{
1009    "function_name": [
1010      {{"param1": value, "param2": value}},
1011      {{"param1": value2, "param2": value3}}
1012    ]
1013  }}
1014
1015  REQUIREMENTS:
1016  - Output strictly valid JSON | no text outside JSON.
1017  - Keys must match function names exactly.
1018  """
1019      if cot_flag:
1020          prompt += "\n" + edge_case_cot_prompt()
1021      return prompt
1022
1023  def edge_case_zero_shot_text() -> str:
1024      return """
1025  Generate diverse edge cases directly for the given functions.
      """

  def edge_case_one_shot_text() -> str:
      return """
  Here is an example of valid edge case JSON:
```

```
{
  "divide": [
    {"a": 10, "b": 2},
    {"a": 10, "b": 0}
  ]
}

Now generate edge cases for the provided functions in the same format.
"""

def edge_case_three_shot_text() -> str:
    return """
Here are examples of valid edge case JSON files:

EXAMPLE 1:
{
  "sqrt": [
    {"x": 4},
    {"x": 0},
    {"x": -1}
  ]
}

EXAMPLE 2:
{
  "factorial": [
    {"n": 5},
    {"n": 0},
    {"n": -3}
  ]
}

EXAMPLE 3:
{
  "substring": [
    {"text": "hello", "start": 1, "end": 3},
    {"text": "hello", "start": -1, "end": 2}
  ]
}

Now generate edge cases for the provided functions in the same JSON format.
"""

def edge_case_cot_prompt() -> str:
    return """
Think step-by-step:
1. Analyze each function signature.
2. Identify normal, boundary, extreme, and invalid input cases.
3. Ensure coverage of exceptions, corner cases, and unusual parameter combinations.
4. Then output ONLY the final JSON with those cases.
"""
```

## B.4    BASELINES UNIT TEST GENERATION PROMPT

### B.4.1    BASELINES LLM UNIT TEST GENERATION SYSTEM PROMPT

```
def edge_cases_to_unittest_system_prompt():
    return """
```

```
    You are an expert Python test generator.
    Your task is to convert the given edge cases
    **and doctest/typical examples** into pytest unit tests.

    RULES:
    1. Output ONLY valid Python 3.11 code | no markdown,
    no explanations, no extra text.
    2. Use EXACTLY 4 spaces per indentation level (no tabs).
    3. All parentheses, brackets, and braces must be balanced.
    4. Import only pytest and built-ins if needed | no extra imports.
    5. Each edge case must become one complete pytest test function.
    6. Test names must follow: test_<function>_<short_scenario>.
    7. Use literals exactly as shown
    (Ellipsis → ..., Infinity → float("inf"), etc.).
    8. Function parameters and variables MUST be valid Python identifiers:
        - Must start with a letter or underscore
        - May contain letters, numbers, or underscores
        - Must NOT start with a digit (incorrect: "3_14" → correct: "val_3_14")
    8a. If the edge case uses unclear or undefined variables
    (e.g., threshold_3_14, Array_1000_0):
        - Replace them with safe, concrete Python literals:
            - Numbers: 0, 1, 3.14
            - Lists: [], [0], [None] as appropriate
            - Strings: '', 'example'
            - Objects: None
    9. Edge case handling:
        - {"input": {...}, "expected": X} → assert function output == X
        - {"input": {...}, "raises": "ExceptionType"}
        → wrap call in pytest.raises(ExceptionType)
        - {"input": {...}} only → just call the function
    9b. For **normal/typical inputs** (including doctests),
    generate pytest functions with **assert statements** for expected results.
    10. Avoid duplicates: if multiple edge cases are semantically identical,
    merge them into one test function.
    11. Every generated test file must pass a syntax check:
        `python -m py_compile generated_tests.py`
    12. Mentally simulate importing and running the file to confirm:
        - All tests execute without NameError, TypeError, SyntaxError,
        or undefined variables.
    13. Always include at least one test for **valid input with assert**,
    even if edge cases exist.
    14. Convert all doctest-style examples (>>> lines)
    into pytest assert statements.
    15. Do NOT invent new literals or variable names;
    always use safe defaults if input is unclear.

    DO NOT OUTPUT ANYTHING OTHER THAN THE TEST CODE.
    """
```

## C  EXAMPLE UNIT TEST FILE GENERATION

To illustrate the workflow of our framework, we provide a concrete example drawn from Django's ORM internals. The source code (Figure 5) contains helper classes and functions that are invoked when constructing SQL queries.

From these source files, our system automatically generates corresponding unit test files. The generated tests (Figure 6) are designed to cover key execution paths and boundary conditions while consisting of runnable test cases.

The source code and the generated unit test file are shortened and simplified for clarity, however, they retains the essential semantics for demonstrating unit test generation.

## C.1 SOURCE FILE

Figure 5 [TOP] shows the definition of the `Q` class. This class is a core building block for query construction: it stores conditions in the `children` attribute, tracks the logical connector (`AND`, `OR`), and exposes the `_combine` method to merge query fragments. The `_combine` method ensures type-safety by restricting merges to other `Q` objects, handles corner cases such as empty children, and creates a new `Q` object with the combined conditions.

Figure 5[BOTTOM] shows the `FilteredRelation` class. This class represents a relation name with an optional condition. It validates that the relation name is non-empty and assigns a default `Q` object if no condition is provided. The equality operator (`__eq__`) is overridden to allow semantic comparison between two `FilteredRelation` objects based on both the relation name and condition.

## C.2 GENERATED UNIT TESTS

Our framework automatically generates the unit test file targeting the key behaviors of these source classes.

Figure 6[BOTTOM] shows tests for `FilteredRelation`. The tests cover: (i) successful equality when both objects have identical fields; (ii) inequality when relation names differ; (iii) inequality when conditions differ; and (iv) type mismatch where equality is checked against a non-`FilteredRelation` object. These cases validate both the intended semantics of the `__eq__` method and its robustness against invalid inputs.

Figure 6[TOP] shows tests for the `Q` class. The generated cases systematically explore: (i) combining with an invalid type (triggering a `TypeError`); (ii) combining when one side has no children; (iii) combining when the current object is empty but the other is non-empty; and (iv) combining two non-empty `Q` objects to ensure the resulting object aggregates children correctly and records the connector string. These unit tests directly exercise the control-flow paths in `_combine`, including exception handling and state mutation.

# D RULE-BASED ENGINE: COLD-START

At initialization, our framework requires a mechanism to seed candidate edge cases before any feedback from coverage or mutation testing is available. We implement this *cold-start* stage through a Python-specific rule-based expansion engine. The engine enumerates deterministic variants of the input state across several dimensions:

- **Numeric values:** Expansion covers boundary conditions such as zero, $\pm 1$, extreme integers ($2^{31}-1$, $2^{63}-1$), large/small floats (e.g., $10^{10}$, $10^{-10}$), infinities, NaNs, and very large arbitrary-precision integers.

- **Strings:** Variants include empty and whitespace strings, boolean-like and number-like encodings, path traversal patterns, injection-style payloads, long Unicode/emoji sequences, and control characters.

- **Lists:** Cases include empty lists, singleton lists, very long lists with repeated values, reversed lists, lists containing `NaN` or `Inf`, and deeply nested structures.

- **Dictionaries:** Variants are created with empty values, `None`-filled keys, or problematic/reserved keys (e.g., `__class__`, whitespace keys, `"True"`).

- **Python special values:** Serializable forms of special objects (e.g., `None`, booleans, empty containers, long strings, lists of `None`) provide coverage of unusual runtime behaviors.

- **Exception triggers:** Values known to raise errors in Python (division by zero, invalid encodings, null-byte strings, memory-exhausting lists) are injected to surface robustness gaps early.

```python
class Q:
    def __init__(self, **kwargs):
        self.children = list(kwargs.items())
        self.connector = None

    def _combine(self, other, conn):
        if not isinstance(other, Q):
            raise TypeError("Can only combine Q objects")

        if not self.children:
            return other
        if not other.children:
            return self

        obj = Q()
        obj.connector = conn
        obj.children = self.children + other.children
        return obj
```

```python
class FilteredRelation:
    def __init__(self, relation_name, condition=None):
        if not relation_name:
            raise ValueError("relation_name must not be empty")
        self.relation_name = relation_name
        self.condition = condition or Q()

    def __eq__(self, other):
        if not isinstance(other, FilteredRelation):
            return False
        return (
            self.relation_name == other.relation_name
            and self.condition == other.condition
        )
```

Figure 5: Simplified excerpt from Django ORM internals

The expansion process is designed to remain JSON-serializable and reproducible, ensuring compatibility with our execution and logging infrastructure. While each individual rule is simple, together they provide broad initial coverage of Python-specific failure modes. This makes the cold-start stage non-trivial: even before iterative search begins, the actor is seeded with high-value candidates that often resolve a substantial fraction of problems, as shown in our HumanEval results (Section 5.1).

# E    USE OF LARGE LANGUAGE MODELS

We employed a large language model (ChatGPT) in a limited capacity to refine the writing of this manuscript. The model's use was restricted to stylistic improvements such as clarity and conciseness. All scientific contributions—including the conception of ideas and algorithms, design of methods, and execution of experiments—were the sole work of the authors.

```python
def test_q_combine_invalid_other_type():
    q_obj = Q()
    with pytest.raises(TypeError):
        q_obj._combine(other="\x00\x01\x02", conn="test_conn")

def test_q_combine_empty_other():
    q_obj = Q()
    result = q_obj._combine(other=Q(), conn="test_conn")
    assert isinstance(result, Q)

def test_q_combine_empty_self():
    q_obj = Q()
    other_obj = Q(some_field=True)
    result = q_obj._combine(other=other_obj, conn="test_conn")
    assert isinstance(result, Q)
    assert result.children == other_obj.children

def test_q_combine_both_non_empty():
    q_obj = Q(some_field=True)
    other_obj = Q(another_field=False)
    result = q_obj._combine(other=other_obj, conn="test_conn")
    assert isinstance(result, Q)
    assert len(result.children) == 2
    assert result.connector == "test_conn"
```

```python
def test_filteredrelation_eq_success():
    obj1 = FilteredRelation(relation_name="valid_relation")
    obj2 = FilteredRelation(relation_name="valid_relation")
    assert obj1 == obj2

def test_filteredrelation_eq_different_relation_name():
    obj1 = FilteredRelation(relation_name="relation1")
    obj2 = FilteredRelation(relation_name="relation2")
    assert obj1 != obj2

def test_filteredrelation_eq_different_condition():
    obj1 = FilteredRelation(relation_name="valid_relation", condition=Q(some_field=True))
    obj2 = FilteredRelation(relation_name="valid_relation", condition=Q(some_field=False))
    assert obj1 != obj2

def test_filteredrelation_eq_not_filteredrelation():
    obj = FilteredRelation(relation_name="valid_relation")
    assert obj != 42
```

Figure 6: Generated unit test file corresponding to Source file

