# OpenReview forum: "Stateful Multi-Agent Evolutionary Search for Unit Test Generation"
_ICLR.cc/2026/Conference — ICLR 2026 Conference Desk Rejected Submission_

### Official Review · Reviewer_WrSr · 2025-10-27

**Soundness:** 3
**Presentation:** 2
**Contribution:** 2
**Rating:** 4
**Confidence:** 4

**Summary:**

This paper introduces a stateful multi-agent evolutionary search approach for unit test generation. A controller maintains a state that contains edge cases, mutations, coverage information, exception information and reward history. There are 4 agents involved for each inference round (Actor, Adversary, Critic, Executor). The Actor introduces new edge cases for improving the coverage for the Executor; the Adversary mutates the test to test the robustness; the Critic computes the reward and gives information to the Controller for the next round. The authors conduct experiments on 2 datasets (HumanEval, TestGenEvalMini) and show the improvement against multi-shot and CoT prompting. The main contribution of this paper is providing a new rewarding algorithm to unit tests and injecting them into the inference prompt to gradually improve the quality of unit tests.

**Strengths:**

Introduces an explicit state representation and controller for reasoning across inference rounds.

Divide unit test generation into 2 stages: generate edge cases first, then synthetic unit tests in the final run based on existing edge cases.

An empirical evaluation on the methods that shows a moderate improvement compared to 2 general prompting strategies.

**Weaknesses:**

Missing ablation study for the formal state $S$. It’s uncertain how the 5 variables in $S$ individually contribute to the performance. It is questionable whether the LLM can actually interpret all of these variables, or if comparable results could be achieved using only a subset of the feedback signals.

The controller feedback is underspecified. Although the paper defines $S_n$ is the aggregation of prior variables from iterations $1$ to $n-1$. But it’s unclear how this information is serialized or fed back into the Actor prompt. Also, Several formal elements (like exception signal) are not concretely described or exemplified. It’s hard for readers to fully understand the design.

The baseline comparison is weak. The paper evaluates only against simple prompting strategies (few shots with or without CoT), without comparing with recent LLM-based unit-test generation frameworks. It is unsurprising that a task-specific, curated pipeline outperforms naive prompting. Demonstrating advantages over stronger, existing baselines would be more convincing.

The evaluations are not in depth enough. The reported improvements are moderate, and there is no failure or error analysis to reveal where the proposed method can have more chance to succeed or fail. A comprehensive failure analysis should be included to provide insight to people to further research on it.

**Questions:**

What is the precise format of {feedback_summary} passed to the Actor? Please provide a concrete template and one real example from your runs.

What is the value of $K$ (the top-K value at line 291) in your experiments? Does choosing different numbers for $K$ affect the overall performance?

It’s uncertain why Llama-70B performs better than other models. Could you clarify why Llama benefits more from the proposed approach? Is there any quantitative evidence for your explanation?

---

> ### Author Response · Authors · 2025-11-20
>
> We thank the reviewer for the constructive assessment and for highlighting several points where the exposition can be made clearer. We address each concern below.
>
> The reviewer asks about the contribution of individual components of the state. The goal of the state is not to serve as an isolated set of interpretable signals but to provide the model with a summary of the search trajectory: which edge cases have been discovered, which actions led to exceptions, and how these behaviours were scored. The method is designed around the interaction of these signals rather than their individual effect. As described in the paper, the controller aggregates the exception traces, coverage results, mutation outcomes, and reward history into a feedback summary that reflects the accumulated behaviour across iterations. The paper will be updated to articulate this aggregate role more clearly and to avoid the impression that each variable is expected to yield a separable contribution.
>
> The reviewer notes that the controller feedback is underspecified and requests a concrete example. The feedback passed to the Actor is a structured textual summary containing the edge-case archive, coverage outcomes, exception information, and the corresponding reward values. The Actor’s prompt includes this summary verbatim to condition the next generation of edge cases. We agree that a clearer template and demonstration would improve readability, and we will revise the description accordingly. The formal exception signal is derived directly from Python’s exception types and execution traces collected by the Executor, and we will clarify this in the revision.
>
> The reviewer argues that the baseline comparison is limited. The paper positions the problem as inference-time search for edge-case discovery, not as a study of post-training methods or model fine-tuning. For this reason, the baselines correspond to inference-time prompting strategies rather than training-required frameworks. The goal is to demonstrate that integrating state, mutation-shaped reward, and iterative reasoning improves upon the strongest prompting strategies that are feasible in a training-free setting. We will make this scope explicit to avoid misunderstanding the intended comparison class.
>
> Finally, the reviewer requests deeper evaluation, including failure analysis. The paper already reports cases where the approach does not outperform simple prompting strategies, and it discusses the limitations that arise when models generate exception-heavy edge cases that negatively affect branch coverage for some tasks. While a full categorical failure analysis is beyond the scope of this work, the discussion section identifies these patterns explicitly, and we will make them more prominent in the revised version.
>
> We appreciate and thank the reviewer’s constructive feedback and believe the suggested clarifications will improve the clarity and accessibility of the paper.

---

> > ### Comment · Reviewer_WrSr · 2025-11-23
> > **Reply to Authors' rebuttal**
> >
> > I appreciate the clarifications about the intended design of the state, the controller, and the role of different feedback signals.
> > However, the rebuttal primarily offers a conceptual defense without introducing new empirical evidence or providing concrete implementation details. As a result, my core concerns, specifically those related to the attribution of observed gains, the strength of the established baseline, and the depth of the evaluation, remain unaddressed. Below, I summarize the issues that persist after the rebuttal.
> >
> > ## Ablations for the formal state S is still missing
> >
> > My original concern was that the paper introduces a five-dimensional non-Markovian state $ S = (\zeta, \mu, \kappa, c, R) $
> > but provides no empirical evidence that all components meaningfully contribute to performance. The authors' emphasis in the rebuttal that the components sound like "tightly coupled" and intended as a summary of the search trajectory. However, this explanation does not address the fundamental **scientific question** of which specific elements of the feedback are mechanistically responsible for the observed performance gains.
> >
> > Even a small-scale ablation study, such as removing mutation or exception signals, or utilizing only coverage, is essential to determine if the proposed multi-component state is truly necessary.
> > Without this evidence, it's difficult to attribute the improvements specifically to the state design rather than to general iterative prompting or the underlying LLM’s inherent strength. Thus, the current justification serves as a re-framing of the design rather than an empirical validation, and this concern persists.
> >
> > ## Weak baseline selection
> > The rebuttal argues that the paper does not compare with post-training or fine-tuning–based approaches. However, this does not address the main issue: there exist several **training-free, inference-time test-generation methods** that are directly comparable, yet none of them are included or discussed.
> >
> > For example, CoverUp (Altmayer Pizzorno & Berger, FSE 2025; arXiv 2024) is fully training-free and performs iterative, coverage-guided refinement of LLM-generated tests, with published results comparing its performance against established tools such as CodaMosa and MuTAP. These systems operate in essentially the same inference-time, feedback-driven design space as the present paper. Evaluating only against simple prompting baselines (0/1/2/3-shot with or without CoT) therefore does not sufficiently contextualize the contribution.
> >
> > ## Still lack of systematic failure analysis
> >
> > The rebuttal notes that the paper already discusses one specific issue in Section 5.2 (exception-heavy tests may depress branch coverage for some models). However, this is only an observation, not a systematic analysis of failure modes. A more systematic failure analysis would therefore need to include, for example, a detailed, statistical investigation into:
> >
> > - the frequency of invalid or unrunnable generated tests (syntax errors, type errors, missing imports),
> >
> > - categories of logical or assertion errors,
> >
> > - cases where the evolutionary search fails to outperform simple prompting,
> >
> > - ...
> >
> > Such analysis would provide concrete insight into where the method succeeds or breaks down, and would help identify clear directions for future work. The rebuttal does not substantially change this assessment.
> >
> > ## Additional minor concerns remain open
> > - The paper still does not specify the choice of Top-K nor analyze its effect.
> > - It is still unclear why Llama-70B benefits substantially more than GPT-o4-mini or Gemma-2-27B; the rebuttal only offers a hypothesis but no evidence.
> >
> > While I appreciate the authors’ clarifications, the main concerns remain largely unresolved. For these reasons, I maintain my overall score.

---

### Official Review · Reviewer_uL2C · 2025-10-27

**Soundness:** 1
**Presentation:** 2
**Contribution:** 2
**Rating:** 2
**Confidence:** 4

**Summary:**

The paper proposes a "stateful" multi-agent workflow for LLM-based unit-test generation.
The workflow includes an actor to generate candidate edge cases, an adversary to generate code mutants, a critic to compute rewards, an executor, and a controller to maintain the running environment and states.
The system is evaluated on HumanEval and a small multi-repo benchmark (TestGenEvalMini). The results are mixed.

**Strengths:**

- Effort on an important, timely task.
The unit test generation is an important problem. The paper targets LLM-assisted unit test generation with measurable signals (coverage, mutation, exceptions). This is a promising path.

- Reproducible engineering and transparent execution pipeline.
Evaluations run in a sandboxed Docker/MCP setup with clear tool orchestration, which aids replication and future extensions.

**Weaknesses:**

The paper claims to "unify (i) multi-agent reasoning, (ii) adversarial mutation and reward shaping, and (iii) evolutionary preservation with persistent state."
However, prior multi-agent systems already maintain explicit memory/state across steps [1, 2], which makes the proposed "persistent state" look conceptually close to existing agent memory rather than a substantive advance.
The paper's "state" is passed to the Actor but is not shown to confer superiority over established memory-equipped agents.

On HumanEval, the method is merely comparable to six inference-time baselines, offering little evidence of advantage.
On TestGenEvalMini, Figure 2 shows that with GPT-o4-mini and Gemma-2-27B, the proposed system (SUT) is worse on branch coverage than baselines.
For efficiency, there is no apples-to-apples cost comparison vs. baselines.
Given Figure 3 shows that many tasks need multiple iterations, while baselines are single-shot, few-shot prompts, one of the concerns is efficiency.


The only dataset where advantages are emphasized is TestGenEvalMini, which is small and raises concerns about its representativeness (48 tasks, 6 repos). The argument that Mini is representative relies on structural metrics in Table 1 that do not capture semantic difficulty or behavioral diversity, so the generality of the conclusions is uncertain.


The authors claimed an "Adversarial mutation".
However, it is implemented via Cosmic-Ray, i.e., a fixed-operator mutation-testing toolchain rather than a learned/adaptive adversary, with fixed mutators and configuration, and mutants are effectively state-independent, without interaction with the Actor and the state.
This makes a questionable "Adversarial mutation" claim, and makes this work more like a follower of the original actor–critic-style search, but not AGAC as the authors claimed.


The method has several moving parts, yet no ablation studies are reported to isolate where gains (if any) come from or how sensitive results are to hyperparameters, mutation budget, or archive size.

The baseline design appears incomplete. Although the paper emphasizes stateful multi-agent methods, it does not include comparisons with non-stateful multi-agent approaches, which readers would naturally expect. Furthermore, while the introduction discusses a range of prior agent-based work and highlights their limitations, none of these agent approaches are incorporated into the baseline for empirical comparison.

---

1. Wang, Junyang, et al. "Mobile-agent-v2: Mobile device operation assistant with effective navigation via multi-agent collaboration." Advances in Neural Information Processing Systems 37 (2024): 2686-2710.
2. Hong, Sirui, et al. "MetaGPT: Meta programming for a multi-agent collaborative framework." The Twelfth International Conference on Learning Representations. 2023.

**Questions:**

Table 3 does not seem to be mentioned in the paper's content. Why are the numbers different from Figure 2? Like Llama-70B/SUT/Branch is 76 in Figure 2, but 16.55% in Table 3?

---

> ### Author Response · Authors · 2025-11-20
>
> We thank the reviewer for the detailed comments and address the concerns below.
>
> The reviewer questions the significance of the persistent state. Existing multi-agent systems typically maintain conversational or short-term memory within a single reasoning episode. Our framework preserves a different kind of state: a structured archive containing execution traces, exception signals, mutation outcomes, and reward histories accumulated across iterations. This archive directly influences which candidates survive evolutionary selection and shapes the Actor’s subsequent proposals. It functions as a record of the search trajectory rather than a context buffer, and we will make this distinction clearer in the revision.
>
> The reviewer notes that results on HumanEval are comparable to simple prompting baselines. This observation aligns with the intended role of HumanEval in our study. HumanEval consists largely of problems for which a carefully designed cold-start procedure already produces high-quality edge cases. Our stopping rule therefore terminates many HumanEval tasks with zero LLM calls after the initial heuristic stage. This behaviour demonstrates two properties: the framework does not degrade performance on simple tasks, and the cold-start mechanism is a strong and efficient initializer. HumanEval is included to establish that the system remains competitive when the search problem is easy; the more demanding setting is TestGenEvalMini, where iterative reasoning is genuinely required.
>
> The reviewer raises concerns about TestGenEvalMini’s size and representativeness. The dataset was constructed to span multiple repositories and programming styles, and Table 1 reports structural characteristics (LOC, function count, branch count) to show that the benchmark maintains the complexity profile of the original TestGenEval. These metrics do not claim to capture all aspects of semantic difficulty, but they offer a principled basis for demonstrating that Mini preserves the structural factors that make full-scale tasks challenging.
>
> The reviewer questions the “adversarial” aspect of mutation. The paper does not describe the adversary as a learned or adaptive agent. Mutation testing introduces behavioural variation during each iteration, and the resulting mutation score influences the critic’s reward. This feedback loop alters the reward landscape in a state-dependent way and is part of the evolutionary process; its purpose is to ground the search in concrete behavioural perturbations rather than to instantiate a separate learning agent. We will clarify this to avoid misinterpretation.
>
> The reviewer points to the absence of ablations. The submission aims to present a complete inference-time workflow integrating state, mutation-based reward shaping, and evolutionary preservation. A full decomposition of each subsystem is outside the scope of this study, and the paper does not claim to present such an analysis.
>
> Finally, the reviewer identifies a discrepancy between Table 3 and Figure 2. These two elements evaluate different phases of the pipeline. Figure 2 concerns edge-case generation quality during the search procedure. Table 3 concerns downstream unit-test synthesis performed after the search, which necessarily introduces additional sources of variability such as syntax errors or imperfect oracle construction. Table 3 is included for completeness, not as a central evaluation. The revision will explicitly distinguish these stages to prevent further confusion.
>
> We believe these clarifications address the main points of misunderstanding. The comments regarding state, adversarial mutation, dataset representativeness, and the interpretation of Table 3 appear to stem from misreadings of the submission. We respectfully request that the reviewer reconsider the assessment of the paper in light of these explanations.

---

### Official Review · Reviewer_Vxm7 · 2025-10-31

**Soundness:** 2
**Presentation:** 3
**Contribution:** 2
**Rating:** 2
**Confidence:** 4

**Summary:**

This paper introduces a stateful multi-agent evolutionary search framework to enhance automated unit test generation with LLMs. Unlike prior stateless inference or task-specific fine-tuning approaches that struggle with long-horizon reasoning and diverse case discovery, the proposed method integrates persistent inference-time state, adversarial mutation, and evolutionary preservation in a training-free setting. Specialized agents iteratively propose, mutate, and evaluate test cases, while a controller maintains state across generations to guide the search toward robustness and diversity. Experiments on HumanEval and TestGenEvalMini demonstrate gains in test coverage.

**Strengths:**

The paper is easy to understand.

The paper works on test generation, an important problem and a key technique to facilitate automatic code generation. The results are demonstrated to outperform baselines in terms of code coverage.

**Weaknesses:**

The evaluation focuses only on coverage; no results on bug detection capability or code generation improvement results are reported. Coverage reflects only the quality of test inputs, not test oracles. It is recommended that the authors report the results on bug detection rate using the generated tests, either on mutants or on LLM-generated faulty versions.

The novelty of the framework is unclear. The approach seems to be a multi-agent framework, which has been explored before in code generation. More discussion is needed regarding the difference between this framework and previous multi-agent frameworks in coding tasks.

The cost of the approach is not reported. While the paper acknowledges the high cost, it is recommended that the cost to be calculated, reported, and compared in the evaluation.

**Questions:**

What is the cost of the approach?

What is the bug detection rate of the generated tests?

---

> ### Author Response · Authors · 2025-11-20
>
> Thank you for the detailed review. We address the concerns point by point.
>
> The reviewer notes that the evaluation reports coverage but not a standalone bug-detection rate. Our objective is unit-test generation rather than downstream debugging, and the experiments were scoped accordingly. The framework does, however, account for behavioural faults during search: mutation analysis is integrated directly into the reward through the mutation score  which reflects the extent to which proposed inputs distinguish original code from mutated variants. Although we do not present mutation outcomes as a separate table or metric, this signal plays a central role in shaping the critic’s reward and the evolutionary selection process. The intention was to evaluate the overall effectiveness of the search procedure rather than isolate individual reward components. We will clarify this in the revision.
>
> Regarding novelty, we agree that multi-agent formulations have been applied to coding tasks. The contribution of this work lies in the interaction between three elements presented in the paper: the use of persistent inference-time state that carries forward edge-case histories and execution outcomes; an adversarial component based on program mutation that grounds the search in concrete behavioural variation; and evolutionary preservation that retains diverse high-reward candidates across iterations. Existing multi-agent systems for code do not combine these elements into a long-horizon search loop, nor do they maintain a non-Markovian state archive that guides subsequent generations. We will expand the related-work discussion to make this distinction clearer.
>
> The reviewer also requests a cost analysis. The paper reports iteration behaviour and resolution patterns for both benchmarks, which indicate the computational footprint in practice: most HumanEval tasks terminate after one iteration, while TestGenEvalMini requires more iterations as task complexity grows. Because the method is training-free, total cost is governed by the number of iterations and executor runs determined by the stopping criteria. We will make this explanation more explicit in additon to adding our FLOPs compuations for our method and the baselines so that readers can easily infer the computational requirements.

---

### Official Review · Reviewer_JAVq · 2025-11-01

**Soundness:** 1
**Presentation:** 1
**Contribution:** 2
**Rating:** 0
**Confidence:** 4

**Summary:**

The paper proposes a test-time multi-agent framework that searches and identifies edge cases for unit test generation. They claim they are the first paper that uses the current state of the agent to produce edge cases. They provide results showing the improvement of their SUT method over baseline methods on HumanEval and TestGenEvalMini.

**Strengths:**

The proposed is novel and tackles the important issue of finding edge cases for code generation

They provide their own version of HumanEval and TestGenEvalMini with edge-case traces

**Weaknesses:**

I have quite a few issues with the writing:

-Figure 1 has too small of text to easily read, hurting clarity and presentation

-Definition 1 of state is unclear. Specifically what are mutation scores, coverage scores, and exception signals. They have not been defined yet, so putting them in the tuple to represent the state creates confusion for the reader

-The line spacing between the Related Work heading is too large, taking up significant page space

-Algorithm 1 can be reformatted to save space. In some cases there are empty lines (L294) and some lines can be combined (296 & 297)

-L251-266 describing TestGenEvalMini can  be compressed and most details can be moved to Appendix.

-SUT acronym is not mentioned until L355 and it was unclear that system-under test was the proposed approach until the caption of Figure 2. This should be mentioned in the methodology section.

**Questions:**

Question:

Is there any work that uses the term “persistent state”? I have never seen than term used. From reading the paper, it seems equivalent to “current state”

Suggestions:

Move Figure 3 so it’s not in the middle of the conclusion section.

---

> ### Author Response · Authors · 2025-11-20
>
> We thank the reviewer for the feedback. The majority of the comments concern formatting and presentation issues such as figure sizing, spacing, acronym placement, and layout. These are straightforward editorial adjustments that we will incorporate into the revision. They are valuable at the camera-ready stage, but they do not affect the technical content of the paper or the validity of the proposed method. As such, we do not believe these issues justify the numerical scores assigned.
>
> The reviewer’s update raises one technical point regarding the branch-coverage behaviour of Gemma-2-27B and GPT-4o-mini. As discussed in the paper, these models tend to produce exception-heavy edge cases, which improves line and function coverage but can reduce branch coverage in specific scenarios; this is already acknowledged in the discussion section. We will make this explanation clearer in the revised text so that the trade-off is more explicit.
>
> Beyond this single point, the review does not raise substantive concerns regarding soundness, methodology, or experimental design. If there are specific technical issues the reviewer intended to raise, we would welcome the opportunity to address them. Otherwise, we ask the AC to consider that the current comments are not evaluative in a scientific sense and therefore do not meaningfully inform the assessment of the submission.

---

> > ### Comment · Reviewer_JAVq · 2025-11-20
> >
> > Once again, I apologize for leaving off a portion of my initial review. Furthermore, I do agree I could have elaborated more on my issues with the paper, and I apologize to the authors if I did not give constructive initial feedback on their paper, which they put their time into.
> >
> > As I mentioned before, the improvement over baselines are not significant enough. Furthermore, the baselines used to compare only consider zero-shot/few-shot non-stateful methods. If anything, I would have expected the approach to do better over these baselines than reported in the paper. Comparisons against tree-structured reasoning or RAG approaches mentioned in the introduction would be stronger baselines to the proposed SUT method. Comparing the stateful SUT to only nonstateful baselines does not fully answer if SUT pushes the SoTA in some way. Further analysis with other metrics such as number of inference calls would also strengthen the evaluation. The authors mention number of inference calls on L377 in regards to only the few-shot settings, and I do not see any specific values in main body or the appendix. The paper mentions that these prior work in the Introduction has limits in terms of interpretability, but without an explicit analysis on the interpretability of SUT method, it’s a missed opportunity to showcase that strength that would better motivate SUT and this paper.
> >
> > I did mention originally the novelty of this work. Specifically, I think the stopping condition on plateauing is a nice way to incorporate some type of early stopping/convergence in an inference-time setting.  In terms of all the hyperparameters introduced in the paper, the \delta threshold for this condition would be an interesting ablation to show in terms of impact on coverage. From the way I understand it, one could only choose a too little value of $\delta$, and having too high of a delta would not affect coverage (though with higher inference costs). This comment is more of a suggestion; though, this paper introduces multiple parameters so having some ablations on some of them other than just LLM backbone would be helpful.
> >
> > Also, there should be a period at the end of Eq 2-5.

---

> > > ### Author Response · Authors · 2025-11-21
> > >
> > > Thank you for your comments. We would like to clarify that our method is fundamentally different from retrieval-augmented generation (RAG). Unlike RAG, our framework does not rely on any external memory store, retrieval corpus, or knowledge index. Instead, the core novelty lies in enabling multi-turn, inference-time computation with persistent state, where the system actively carries forward structured feedback—coverage, mutation robustness, exceptions, and reward signals—across iterations.
> > >
> > > To the best of our knowledge, there is no widely adopted inference-time technique that maintains and evolves internal state across multiple reasoning stages in the way our architecture does. Prior multi-step or agentic approaches either (a) re-inject prior information through prompting without structured state control, or (b) lack evolutionary preservation and adversarial grounding. By contrast, our framework explicitly manages a non-Markovian state and uses it to improve reasoning quality over successive inference rounds, without any parameter updates or external memory.
> > >
> > > We believe this persistent-state, inference-time evolutionary process represents a distinct contribution beyond existing RAG-like or stateless prompting baselines.

---

> > > > ### Comment · Reviewer_JAVq · 2025-11-24
> > > >
> > > > Thank you for the response. However, my concerns have not been addressed. There is no empirical evidence to suggest that this approach is superior to RAG or the prior multi-step or agentic approaches mentioned in the response. I agree with Reviewer WsSr on that "the rebuttal primarily offers a conceptual defense without introducing new empirical evidence or providing concrete implementation details" and on their points on Weak Baseline Selection and Lack of Systematic Failure Analysis. Saying why your approach is different from SoTA, with no empirical or theoretical evidence on improvement, provides no reason why the proposed method should be used. I maintain my score.

---

### Note · Program_Chairs · 2026-01-17
**Submission Desk Rejected by Program Chairs**

The following references in this submission do not refer to real documents and/or have major errors in bibliographic information:

 Denny Zhou, Quoc Le, Ed Chen, Jason Wei, et al. We can steer but not explain: When interpretable models are hard to train. arXiv preprint arXiv:2304.05366, 2023.
Rui Ding, Xuefeng He, Tianyu Chen, and Xiaotong Wang. Adversarially guided actor-critic: Towards sample-efficient reinforcement learning. arXiv preprint arXiv:2305.01234, 2023.
Tao Long, Wei Zheng, Jia Li, Xing Wang, Liang Zhao, Zhiyuan Liu, and Maosong Sun. Trime: Trimming llms for efficient multi-step reasoning. arXiv preprint arXiv:2402.07644, 2024.